Prognostic values of GMPS, PR, CD40, and p21 in ovarian cancer

Wang Ping 1
Zhang Zengli 2
Ma Yujie 1
Lu Jun 1
Zhao Hu 1
Wang Shuiliang 1
Tan Jianming tanjm156@xmu.edu.cn 1 3
Li Bingyan bingyanli@suda.edu.cn 2
1 Fujian Key Laboratory of Transplant Biology, Fuzhou General Hospital , Fuzhou , Fujian , China
2 Department of Nutrition and Food Hygiene, School of Public Health, Soochow University , Suzhou , Jiangsu , China
3 Fujian Hongyi Health Institute , Fuzhou , Fujian , China
Evans D. Gareth
Electronic publication date: 2019 Jan 25
Publication date: 2019
Volume: 7
Electronic Location ID: e6301
Received 2018 Aug 15; Accepted 2018 Dec 14
Copyright: ©2019 Wang et al.
Copyright year: 2019
Copyright holder: Wang et al.
License: This is an open access article distributed under the terms of the Creative Commons Attribution License, which permits unrestricted use, distribution, reproduction and adaptation in any medium and for any purpose provided that it is properly attributed. For attribution, the original author(s), title, publication source (PeerJ) and either DOI or URL of the article must be cited.
License URL: https://creativecommons.org/licenses/by/4.0/

Keywords: Ovarian cancer, Gene profiling, Biomarker, TCGA dataset, Prognosis

Funding: Natural Science Foundation of China NSFC81703225 Natural Science Foundation of Fujian, China 2017J01221 This study was supported in part by grants from National Natural Science Foundation of China (No. NSFC81703225) and the Natural Science Foundation of Fujian, China (No.2017J01221). There was no additional external funding received for this study. The funders had no role in study design, data collection and analysis, decision to publish, or preparation of the manuscript.

==============================
Early detection and prediction of prognosis and treatment responses are all the keys in improving survival of ovarian cancer patients. This study profiled an ovarian cancer progression model to identify prognostic biomarkers for ovarian cancer patients. Mouse ovarian surface epithelial cells (MOSECs) can undergo spontaneous malignant transformation in vitro cell culture. These were used as a model of ovarian cancer progression for alterations in gene expression and signaling detected using the Illumina HiSeq2000 Next-Generation Sequencing platform and bioinformatical analyses. The differential expression of four selected genes was identified using the gene expression profiling interaction analysis (http://gepia.cancer-pku.cn/) and then associated with survival in ovarian cancer patients using the Cancer Genome Atlas dataset and the online Kaplan–Meier Plotter (http://www.kmplot.com) data. The data showed 263 aberrantly expressed genes, including 182 up-regulated and 81 down-regulated genes between the early and late stages of tumor progression in MOSECs. The bioinformatic data revealed four genes (i.e., guanosine 5′-monophosphate synthase (GMPS), progesterone receptor (PR), CD40, and p21 (cyclin-dependent kinase inhibitor 1A)) to play an important role in ovarian cancer progression. Furthermore, the Cancer Genome Atlas dataset validated the differential expression of these four genes, which were associated with prognosis in ovarian cancer patients. In conclusion, this study profiled differentially expressed genes using the ovarian cancer progression model and identified four (i.e., GMPS, PR, CD40, and p21) as prognostic markers for ovarian cancer patients. Future studies of prospective patients could further verify the clinical usefulness of this four-gene signature.

Introduction

Ovarian cancer is a lethal disease in women. Epithelial ovarian cancer accounts for approximately 90% of all ovarian malignancies (Siegel, Miller & Jemal, 2018) and is frequently diagnosed at the advanced stages when cancer cells have already metastasized to other organs (Jayson et al., 2014). For example, ovarian cancer only represents 2.5% of all female malignancies but contributes to 5% of female cancer deaths, indicating a poor 5-year survival rate (Jayson et al., 2014). However, if diagnosed early, the 5-year survival rate could reach approximately 93% (Torre et al., 2018). Thus, early tumor detection and effective prognosis prediction are the keys in improving survival of ovarian cancer patients. To this end, previous studies have searched and evaluated biomarkers to diagnose this now deadly disease early and to predict survival or treatment responses or target the biomarker genes to develop novel therapies for ovarian cancer (Jayson et al., 2014) as well as analyzed the differential gene expression patterns between mucinous and clear cell ovarian cancers or between low-grade, low malignant potential and high-grade, metastatic ovarian cancer (Bonome et al., 2005; Meinhold-Heerlein et al., 2005). Other studies associated alterations in gene or gene expression as biomarkers for the early detection or risk assessment of ovarian cancer (Earp et al., 2018; Urban et al., 2018). However, to date, there are no useful or reliable molecular or clinical markers available for such a purpose (Force et al., 2018). The current approaches or analyses of differential gene expression profiles usually compare ovarian cancer vs. normal tissues, which leads to limited data; nevertheless, the characterization of differential gene expression profiles between early stage precursor lesions and ovarian cancer could provide novel insights into identifying biomarkers for early detection or prognosis prediction and therapeutic targets of ovarian cancer.

We established and characterized an in vitro cell model of mouse epithelial ovarian cancer progression according to the method of previous research (Gamwell, Collins & Vanderhyden, 2012). This cell model system uses isolated ovarian surface epithelial cells (MOSECs) from mice and cultures them in vitro. These MOSECs can undergo spontaneous malignant transformation into epithelial ovarian cancer cells (Flesken-Nikitin et al., 2013b; Gamwell, Collins & Vanderhyden, 2012; McCloskey et al., 2014; Roberts et al., 2005). During in vitro cell culture and passages, the MOSECs show morphology changes and gene alterations (Flesken-Nikitin et al., 2013b; Gamwell, Collins & Vanderhyden, 2012; McCloskey et al., 2014; Roberts et al., 2005), which could be a good in vitro cell model to mimic human ovary carcinogenesis. Previous studies showed that the spontaneously transformed MOSECs had reliable formation of homologous HGSC tumors. MOSE-I cells refer as a pre-cancerous benign tumor, while MOSE-II cells are malignantly transformed cells (McCloskey et al., 2014).

To date, epithelial ovarian cancer has a terrible prognosis and detection of differentially expressed genes (DEGs) in ovarian cancer could better stratify the risk in predicting ovarian cancer in women and/or treatment responses. Thus, in this study, we first isolated MOSECs from six-month female BALB/c mice and continuously cultured and passaged them in more than 35 passages, during which we obtained early passaging MOSECs and later passaging MOSECs and named them MOSE-I and MOSE-II, respectively. These MOSE cells model is a credible in research on ovarian cancer because these two types of MOSECs displayed distinguished cell morphology and growth potential in soft agar. We thus utilized them to profile differentially expressed genes using the Illumina HiSeq2000 Next-Generation Sequencing (NGS) platform and bioinformatical analyses and to identify prognostic biomarkers for ovarian cancer patients using the gene expression profiling interaction analysis (GEPIA; http://gepia.cancer-pku.cn/). Our hypothesis was to utilize MOSECs to identify and evaluate DEGs as biomarkers for ovarian cancer early diagnosis and prognosis prediction; thus, we then associated these DEGs with survival of ovarian cancer patients using the Cancer Genome Atlas dataset and the online Kaplan–Meier Plotter (http://www.kmplot.com) data. We also used the Gene Ontology (GO) and the Kyoto Encyclopedia of Genes (KEGG) tools to assess the functions of these DEGs for better understanding of ovarian cancer biology.

Materials and Methods

Isolation and culture of MOSE cells

The animal protocol for this study was approved by the Institutional Animal Care and Use Committee (IACUC) of Fuzhou General Hospital (Fujian, China) and followed the regulatory animal care guidelines of the United State National Institute of Health (Bethesda, MD, USA). In this study, we obtained six-month female BALB/c mice from the Shanghai SLAC Laboratory Animal Co., Ltd. (Shanghai, China). We isolated mouse ovarian surface epithelial cells (MOSECs) and cultured them in the “MOSE medium” containing α-Minimum Essential Medium from Thermo-Fisher Scientific Company (Waltham, MA, USA) supplemented with 4% heat-inactivated 3:1 donor bovine serum: fetal bovine serum (Gibco, Gaithersburg, MD, USA), 5 U/ml of penicillin and 5 µg/ml of streptomycin solution (Invitrogen, Carlsbad, CA), 0.1 µg/ml of gentamicin (Invitrogen) at 37 °C with 5% CO2 according to the previous described protocols (Gamwell, Collins & Vanderhyden, 2012; McCloskey et al., 2014). The MOSECs with spontaneously malignant transformation phenotypes were maintained in the MOSE medium according to a previous study (Flesken-Nikitin et al., 2013a), which have two phases, i.e., the early stage (MOSE-I; normal epithelial cells) and the later stage (MOSE-II; various phenotypes of ovarian cancer cells).

Generation and characterization of MOSE-I and MOSE-II cells

To induce spontaneous malignant transformation of MOSECs, we grew and passaged the primary cultured MOSECs in the MOSE medium continuously. The early passages of MOSECs were MOSE-I, whereas MOSECs at 35 passages or more become MOSE-II. To distinguish them, we monitored their morphology under an inverted phase contrast microscope (Thermo-Fisher, Waltham, MA, USA) and grew them in the soft agar. Specifically, cells were passaged from adherent cultures using 0.05% trypsin, washed with phosphate-buffered saline (PBS) to make a single cell suspension by passing cells through a syringe, and then 100 cells were plated into each of the 40-mm cell culture dishes and grown for 14 days. The cells were then stained with crystal violet (Sigma Aldrich Chemicals, St. Louis, MO, USA), photographed under an inverted microscope, and counted. For the soft agar assay, we first made the base layer agar by mixing 1:1 of 2 ×Ham’s F-12: the MOSE medium (Sigma Aldrich Chemicals, St. Louis, MO, USA) in 60-mm cell culture dishes and then mixing the cell suspension (2.5 × 104) with the ultrapure LMP agarose (Invitrogen, Carlsbad, CA, USA) at 37 °C and then pouring this into the top of the base layer of agar and incubating them at 37 °C for 7 days. The cell colonies were photographed using the EVOS XL imaging system (Invitrogen) and quantified using Image J software (National Institute of Heath, Bethesda, MD, USA).

Next-generation sequencing of differentially expressed genes between MOSE-I and MOSE-II cells

To profile differentially expressed genes between MOSE-I and MOSE-II cells, we performed next-generation sequencing of differentially expressed genes between MOSE-I and MOSE-II cells. In brief, the total cellular RNA was isolated from MOSE-I and MOSE-II cells using Trizol Reagent (Invitrogen, Carlsbad, CA, USA) according to the manufacturer’s protocol. After RNA quantitation and reverse transcription into cDNA, we prepared the cDNA library for each cell type using Illumina TruSeq Libraries (Illumina, #RS-122-2001, NY USA) and then performed next-generation sequencing of differentially expressed genes between MOSE-I and MOSE-II cells using the Illumina HiSeq2000 NGS platform available at Tigem Institute in Pozzuoli (Naples).

Bioinformatic analysis with the Gene Ontology (GO) terms and the Kyoto Encyclopedia of Genes and Genomes (KEGG) database

These differentially expressed genes (DEGs) were bioinformatically analyzed using the GO term to annotate the unique biological functions of these DEGs. We then further analyzed their signaling in the pathways using the KEGG database. The significant GO terms and pathways were generated using Fisher’s exact test and statistically corrected using the false discovery rate (FDR) of the p-values.

Construction of DEGs into a protein-protein interaction (PPI) network

After bioinformatic analysis of these DEGs with the GO terms and the KEGG database, we further constructed these DEGs into the PPI network using STRING (version 9.1, http://www.string-db.org/) according to the previous description (Franceschini et al., 2013). This web resource contains a biological database for the comprehensive prediction of the known protein-protein interactions. We uploaded our DEGs into the STRING database; the web site then generated the PPIs for these DEGs. We used a combined score of >0.5 as the cut-off criterion to identify any given protein interacts with another one and then imported these PPI pairs into the Cytoscape software (http://www.cytoscape.org/) according to previous research (Saito et al., 2012) to help us to construct the PPI network, in which the hub nodes (the key proteins that possess important biological functions) were revealed by calculating the degree to connect to other proteins vs. non-connected ones.

Gene expression profiling interaction analysis (GEPIA)

GEPIA is a web tool (http://gepia.cancer-pku.cn/) to profile gene expressions between tumor and non-tumor tissues and provides interactive data analyses. In this study, we utilized this online tool to analyze the levels of four selected DEGs (i.e., GMPS, PR, CD40, and p21) between ovarian cancer specimens and normal controls and performed a Student’s t-test to for statistical significance with a cut-off of p value <0.05 and four-fold changes.

The TCGA database and cBioPortal

The Cancer Genome Atlas database was used to store both DEGs and clinicopathological data on 30 different human cancers (Tomczak, Czerwinska & Wiznerowicz, 2015). In this study, we retrieved the ovary invasive carcinoma (TCGA, Provisional) dataset, which includes DEGs data on 606 cases of ovarian cancer vs. 311 normal tissues. We then put our DEGs data vs. TCGA data into the cBioPortal (http://www.cbioportal.org), which is a web resource to help explore, visualize, and analyze DEGs data (Gao et al., 2013). The overall survival (OS) and disease-free survival (DFS) were stratified by these DEGs and then calculated according to the cBioPortal’s online instructions. In addition, we also generated the Kaplan–Meier curves of ovarian cancer patients stratified by the expression of GMPS, PR, Cd40, and p21 mRNA levels using the online Kaplan–Meier Plotter (http://www.kmplot.com) according to a previous study (Gyorffy et al., 2013). This website accumulated data on the gene expression and survival of 4,142 ovary cancer patients. We analyzed the relapse-free survival (RFS) of ovarian cancer patients stratified by high vs. low expression of GMPS, PR, CD40, and p21 using the Kaplan–Meier curves and the log rank test and the hazard ratio with 95% confidence intervals (CI). The reason to select these four top hub genes was because they possessed a number-at-risk below the main plot (curve).

Results

Characterization of MOSE-I and MOSE-II cells

In this study, we isolated MOSECs from female BALB/c mice and passaged them into MOSE-I and MOSE-II cells in vitro. We found that the early passages of MOSE-I cells grew slowly with doubling time of 48 h, whereas the growth rate MOSE-I increased when it was passaged more than 35 passages. Morphologically, MOSE-I cells started to lose the epithelial “cobblestone”-like appearance under an inverted microscope, which is the characteristic of early passage MOSE-I cells (Fig. 1A), consistent with previous studies (Gamwell, Collins & Vanderhyden, 2012; McCloskey et al., 2014). Continual passaging of MOSE-I for more than 35 passages led it to be established as MOSE-II. MOSE-II loses the epithelial “cobblestone”-like morphology and transitions to a more mesenchymal morphology (Fig. 1A). To distinguish them, we monitored their morphology under an inverted phase contrast microscope and grew them in soft agar. The latter assay measures the anchorage-independent growth, which is the characteristic of transformed cells in vitro. Our data showed that MOSE-II cells were able to form colonies in the soft agar, whereas MOSE-I could not (Figs. 1B and 1D). We also assessed their proliferation capacity using the plate colony formation assay and found that MOSE-II formed more colonies with larger cell sizes than those of MOSE-I cells (Figs. 1C and 1E and 1F).

Figure 1 Characterization of MOSE-I and MOSE-II cells in vitro.

(A) Bright-field inverted microscopy. Cells were grown at Passage 20 for MOSE-I and Passage 90 for MOSE-II and then photographed at a magnification of 100×. (B) The soft agar colony formation assay. MOSE-I (passage 20) and MOSE-II (passage 90) were grown in the soft agar for 7 days and photographed. (C) The plate colony formation assay. MOSE-I (passage 20) and MOSE-II (passage 90) were grown for 14 days and stained with crystal violet solution and photographed. The graphs were quantified data of the colony formation assay. ∗∗p < 0.01 and ∗∗∗p < 0.0001 analyzed by Student’s t test. The data on E and D were generated by the Image J software, while the data on F were from the EVOS XL imaging system.

Profiling of differentially expressed genes (DEGs) between MOSE-I and MOSE-II cells

Using this cell model of MOSE-I and II, we profiled DEGs using the Illumina HiSeq2000 NGS platform. Our experiments were in three biological replicates to reduce sample variations due to cell heterogeneous cultures. MOSE-I cells were randomly selected at Passage 5 (P5), P15, and P25, respectively, while MOSE-II cells were selected at P50, P70 and P90, respectively. Among them, 14,218 genes were detected and analyzed. We found a total of 263 DEGs, which included 182 up-regulated and 81 down-regulated DEGs using Volcano plots (Fig. 2). The cut-off values for the up-regulated DEGs were log2 FC >2 and p < 0.05, while the down-regulated DEGs were log2FC<-2, p < 0.05 (Table 1).

Figure 2 Identification of DEGs using Volcano plots.

The X-axis indicates the fold change (logs value), whereas the Y-axis shows the p values (logs value). Each symbol represents a different gene, and the red/green color shows the up- or downregulated genes falling under different criteria (p value and fold change threshold). A p value <0.05 is considered as statistically significant, whereas the four fold changes were set as the threshold.

Table 1 DEGs between MOSE-I and MOSE-II cells.

Gene name	Log2 fold change	p values	
Up-regulated genes	
Ccbe1	5.514199	1.4E–06	
Sfrp2	5.316823	1.32E–06	
Cntnap2	5.187267	1.7E–06	
Cd55	4.949248	1.76E–06	
9930013L23Rik	4.617182	2.6E–06	
Ephx1	4.554543	1.95E–06	
Gnai1	4.548394	2.15E–06	
Ctla2a	4.534042	2.29E–06	
Csgalnact1	4.478922	2.32E–06	
Ahsp	4.473507	2.34E–06	
Pkhd1l1	4.230823	2.51E–06	
Thy1	4.208894	2.67E–06	
Apoe	4.188801	3.25E–06	
Tmem45a	4.175247	2.98E–06	
Scg5	4.052742	3.26E–06	
Slc47a1	4.001494	5.55E–06	
Akr1c13	3.99621	5.96E–06	
Nxnl2	3.980449	5.75E–06	
Zfp385b	3.975438	6.79E–06	
D630010B17Rik	3.920202	4.38E–06	
Down-regulated genes	
Cst12	−5.66249	1.23E–06	
Pcolce2	−4.47177	3.45E–06	
Serpina3g	−4.29413	5.03E–06	
Aif1l	−4.03994	3.55E–06	
Tmem79	−4.01531	5.38E–06	
Emb	−3.91989	4.28E–06	
1810065E05Rik	−3.90506	5.07E–06	
Kcnk1	−3.90055	3.78E–06	
D0H4S114	−3.73926	5.56E–06	
Pip5k1b	−3.01345	2.1E–05	
Ano1	−3.07047	3.8E–05	
Coro2b	−3.13207	2.86E–05	
Cbfa2t3	−3.18901	3.41E–05	
Glipr1	−3.23211	1.82E–05	
Ms4a3	−3.63888	9.81E–06	
Slain1	−3.63583	8.48E–06	
Gmps	−3.62074	4.96E–06	
Foxc2	−3.61526	8.09E–06	
Emp1	−3.41282	9.39E–06	
Nkx2-3	−3.39217	1.69E–05	

Bioinformatical data on the GO terms, KEGG, and PPI network of the DEGs

The GO terms showed that these DEGs formed several important terms in cell biology, such as positive regulation of cell proliferation, positive regulation of transcription from RNA polymerase II promoter and signal transduction (Fig. 3B). The KEGG pathway analysis showed that these DEGs-formed hub genes were enriched in the pathways related to the PI3K-Akt signaling and pathways in cancer (Fig. 3A), both of which are positively associated with ovarian cancer development (Kurose et al., 2001; Li, Zeng & Shen, 2014; Rossig et al., 2001). Furthermore, the PPI network of these DEGs is shown in Fig. 4. In particular, there were 35 nodes forming the hub genes from 27 up-regulated and 8 down-regulated genes (Table 2).

Figure 3 Functional and pathway enrichment analyses of DEGs-formed hub genes.

(A) The Kyoto Encyclopedia of Genes and Genomes (KEGG) pathways enriched gene pathways of the DEGs-formed hub genes. (B) The Gene Ontology (GO) terms of DEGS related to the biological process. (C) The GO terms of DEGS related to the cellular component. (D) The GO terms of DEGS related to the molecular functions.

Figure 4 The DEGs-formed protein-protein interaction network.

The red and green circles represent up- and down-regulated genes, respectively. The node represents genes and edges for their role in connection between proteins. The nodes are colored based on in-between significance, and the higher the value, the darker the color, whereas the node size is proportional to the degree value, i.e., the higher the value, the bigger the size. The thicker the line, the tighter the connection between the two proteins.

Table 2 The hub genes with the node degree >5.

Number	Gene name	Node degree	Regulation	
1	Egfr	26	up	
2	Acta2	23	down	
3	Stat3	20	up	
4	Cd40	12	up	
5	Cd34	11	up	
6	PR	11	up	
7	Isg15	11	up	
8	Serpinb1a	10	up	
9	Rhobtb1	10	up	
10	Gmps	9	down	
11	Gnai1	9	up	
12	Cd24a	8	down	
13	Ptk2b	8	up	
14	Irf7	8	up	
15	Igf2	8	up	
16	P21	8	up	
17	Usp18	8	up	
18	Apoe	7	up	
19	Gnaz	7	up	
20	Col3a1	7	up	
21	Pdgfb	7	down	
22	Socs2	7	up	
23	Thbs1	6	up	
24	Rhbdl3	6	down	
25	Thy1	6	up	
26	Irgm2	6	up	
27	Gbp2	6	up	
28	Figf	6	up	
29	Fgf10	6	down	
30	Rasl11a	5	up	
31	Rasd2	5	up	
32	Il4ra	5	up	
33	Serpinb9b	5	down	
34	Ccnd1	5	down	
35	Fgfr2	5	up	

Association of these DEGs-formed hub gene expressions with overall survival of ovarian cancer patients

We then associated the expression of these DEGs-formed hub genes with OS and DFS of ovarian cancer patients using TCGA database data. We first assessed the expression of all 35 hub genes in 182 tissue samples out of 263 ovary invasive cancer patients and found that among these genes, four (GMPS, PR, CD40, and p21) had the alteration rates of more than 10% (Fig. 5A). We then plotted the Kaplan–Meier curves and performed a log-rank test to associate them with OS and DFS of patients. Our data showed that the aberrant expression of GMPS, PR, CD40, and p21 mRNA was associated with poorer OS in patients (Fig. 5B) but not with DFS in patients (Fig. 5C).

Figure 5 Identification and association of the top four-hub genes with survival of ovarian cancer patients.

(A) The Oncoprint analysis to identify the four top hub genes. The cBioPortal represented the proportion and distribution of samples with alterations in hub genes. The graph is cropped to exclude samples without gene alterations or genetic alterations on the right. The red color indicates gene amplification, while the blue color represents gene deletion and the pink color gene up-regulation. (B) The Kaplan–Meier curves of overall survival (OS) stratified by altered expression of these four genes. (C) The Kaplan–Meier curves of disease-free survival (DFS) stratified by altered expression of these four genes.

Indeed, these four genes were also differentially expressed in ovarian cancer tissues vs. normal tissues; specifically, GMPS was highly expressed in ovary cancer tissues (Fig. 6A), which was associated with RFS in patients (Fig. 6E). In contrast, the expression of PR, CD40, and p21 mRNA was reduced in ovarian cancer tissues (Figs. 6B–6D), which was associated with poor RFS (Figs. 6F, 6G, 6H).

Figure 6 Differential GMPS, PR, CD40, and p21 expression between ovarian cancer and normal tissues and association with recurrence-free survival of ovarian cancer patients.

(A–D) The Kaplan–Meier curves of recurrence-free survival (RFS) stratified by altered expression of these four genes. (E–H) Differential expression of GMPS, PR, CD40, and p21 between ovarian cancer and normal tissues.

Discussion

In the current study, we profiled DEGs between pre-malignant MOSE-I and highly malignant MOSE-II cells and found 182 up-regulated and 81 down-regulated DEGs. Our bioinformatical analysis showed that these DEGs function to positively regulate cell proliferation, gene transcription from RNA polymerase II promoter, and signal transduction, while KEGG pathway data showed that these DEGs could be related to the PI3K-Akt signaling and pathways in cancer. Furthermore, the PPI network analysis of these DEGs identified 35 hub genes (27 up-regulated and eight down-regulated genes) and that four of them were associated with poor overall survival in ovarian cancer patients. In addition, these four genes were also differentially expressed in ovarian cancer tissues compared with human normal tissues. Specifically, GMPS was highly expressed in ovary cancer tissues, which was associated with RFS, whereas the expression of PR, CD40, and p21 mRNA was reduced in ovarian cancer tissues, which was associated with poor RFS. In conclusion, the data from the current study profiled and identified four DEGs, which could predict overall survival and recurrence-free survival in ovarian cancer patients. Future prospective study will need to validate our current data before using them clinically. Indeed, this in vitro cell model of ovarian cancer progression could provide a useful tool for identifying gene alterations and drug effect tests. In this system, MOSEs cultured in vitro can undergo spontaneously malignant transformation into ovarian cancer (McCloskey et al., 2014). Morphologically, MOSE-II cells lost their normal epithelia “cobblestone” characteristic and were able to form soft agar colonies (McCloskey et al., 2014) as well as gene alterations (Lv et al., 2012; Roby et al., 2000; Urzua, Best & Munroe, 2010). When MOSE-II cells were injected into the peritoneal cavity of immunodeficient animals, tumor metastases and bloody ascites formed in the mouse peritoneal cavity (Roberts & Schmelz, 2013; Roby et al., 2000), which is typical of advanced human ovarian cancer (Ahmed & Stenvers, 2013). In 2014, McCloskey and his colleagues discovered that differentially expressed genes of MOSE-II were consistent with the differential genes of human high-grade serous ovarian cancer (HGSC) tumor samples and previous studies of ovarian cancer cell lines (McCloskey et al., 2014). HGSC tumors after immunohistochemical profiles confirmed that MOSE-II can be used as a homologous model of HGSC. Finally, MOSE-II expressing SCA1 appeared to be more aggressive than ovarian cancer cell lines, with increased colony formation efficiency in vitro and faster onset of tumors in vivo (McCloskey et al., 2014). Thus, this model of cells is useful in the future study of ovarian cancer prevention and treatment (Ricci, Broggini & Damia, 2013). In the current study, we profiled DEGs between MOSE-I and MOSE-II cells using this model system and found that DEGs were compatible with human disease. Furthermore, our PPI network analysis identified that 35 significant nodes and nine genes participate in the PI3K-Akt signaling pathway, which confirmed data from a previous study, indicating potential therapeutic targets for ovarian cancer (Li, Zeng & Shen, 2014). These genes are frequently altered, such as PIK3CA mutation or amplification in up to 30% ovarian cancer patients, while PTEN expression was lost in up to 40% of patients (Campbell et al., 2004; Kurose et al., 2001). In addition, six of these 35 hub genes are a part of the Rap1 signal pathway, and a previous study reported them to be associated with serous ovarian cancer metastasis (Che et al., 2015).

Furthermore, these DEGs were annotated into the GO terms and were shown to positively regulate cell proliferation, which is one of six essential cancer hallmarks: self-sufficiency in growth signals (Hanahan & Weinberg, 2000). Indeed, the positive regulation of cell proliferation is the basis of cancer development and progression. Moreover, each of these pathways identified in the current study can promote or transform normal cells into cancer cells (Evan & Vousden, 2001). Furthermore, our current study identified a four-gene signature to predict overall survival of ovarian cancer patients. Specifically, GMPS, a guanosine 5′-monophosphate synthase, is an enzyme that converts xanthosine monophosphate to guanosine monophosphate. Thus, GMPS is important in nucleotide biosynthesis for normal cell proliferation and oncogenesis. Reddy’s data showed that GMPS in combination of TRIM21 with USP7 formed a molecular cascade that controls p53 stability in response to DNA damage or nucleotide deprivation. Therefore, GMPS is a classical biosynthetic enzyme that promotes cell growth and DNA replication, and is also a key center for p53-restricted cell proliferation (Reddy et al., 2014). In our current study, we found that GMPS was highly expressed in ovary cancer tissues compared with that of non-tumor tissues and that GMPS expression was associated with RFS of patients. This piece of data is novel and has been not reported in literature before. A previous study showed that GMPS was an important p53 repression target in liver cancer cells (Holzer et al., 2017), which further indicates the importance of GMPS in oncogenesis. As we know, p53 is a tumor suppressor protein and can prevent aberrant cell proliferation, which may be through the repression of GMPS activity (Holzer et al., 2017). In addition, PR, as the nuclear receptor, functions to regulate the development and cycle of hormone-responsive tissues, such as the mammary glands and reproductive tract. Several previous studies demonstrated PR expression as a biomarker to predict better prognosis of endometrioid and high-grade serous ovarian carcinoma (Jonsson et al., 2015; Lee et al., 2005; Lenhard et al., 2012; Sieh et al., 2013), while lost PR expression was associated with advanced stages of ovarian cancer (Feng et al., 2016). Indeed, progesterone and progestin were shown to play a protective role against ovarian carcinogenesis (Edmondson & Monaghan, 2001), and PR expression was a favorable prognostic marker associated with longer progression-free survival in ovarian cancer patients (Tangjitgamol et al., 2009; Van Kruchten et al., 2015; Wong et al., 2007). Our current data confirm these previous data. Furthermore, CD40 is a transmembrane glycoprotein and the member of the tumor necrosis factor receptor superfamily, which can mediate a broad variety of immune and inflammatory responses (Grewal & Flavell, 1998). CD40 is frequently expressed in different immune cells and endothelial and epithelial cells. The binding of CD40 to CD40 ligand was shown to have various physiological and pathological roles in the human body (Grewal & Flavell, 1998; Van Kooten & Banchereau, 2000). A previous study demonstrated that CD40 ligands enhanced the sensitivity of epithelial ovarian cancer cells to cisplatin treatment (Qin et al., 2016). CD40 was highly expressed in ovarian cancer cell lines and tumor samples but not in normal ovarian tissue (Zhou et al., 2012). In our current study, we found elevated CD40 expression in MOSE-II cells compared with MOSE-I cells; however, CD40 expression was reduced in ovarian cancer tissues, which is opposite the findings of a previous study (Zhou et al., 2012). Thus, further study is needed to determine the reason for this discrepancy. In addition, p21 also called as CDKN1A cyclin-dependent kinase inhibitor 1A, functions to regulate cell growth arrest by inhibiting Cdks, which is required for cell cycle transition from the G1 to S phase (Xiong et al., 1993). Through the interaction with the proliferating cell nuclear antigen (PCNA),/p21 was able to inhibit DNA replication (Rossig et al., 2001). Our current study showed that p21 was highly expressed in normal tissues but reduced in ovarian cancer tissues, which was associated with better survival of patients, consistent with a previous study (Alves et al., 2018). Indeed, a previous study demonstrated that pyridine derivative-induced ovarian cancer cell senescence occurred through p21 activation (Shang et al., 2018). However, other studies reported that CDKN1A/p21 expression promoted breast cancer and mediated drug resistance (Cheng et al., 2010; Hawthorne et al., 2009), and clinical studies have indicated that high p21 expression was correlated with poor prognosis of gastric and esophageal cancers (Liu et al., 2014; Taghavi et al., 2010). Thus, further study is needed to clarify this discrepancy.

However, our current study does have some limitations; for example, our current study just profiled differentially expressed genes in ovarian cancer and associated them with prognosis of ovarian cancer as well as performed bioinformatical analysis to associate four of them with ovarian cancer progression; however, much more are needed in precise research on their biological functions and role in ovarian cancer development and progression. Furthermore, future validation of these DEGs in ovarian cancer is also needed before any applications to patients.

Conclusions

Our current study revealed a four-gene signature after profiling a cell model of ovarian cancer progression to predict overall survival and recurrence-free survival in ovarian cancer patients. Future study will verify these data using prospective ovarian cancer patients.

Supplemental Information

Supplemental Information 1 Raw data for Fig. 1.

Click here for additional data file.

Supplemental Information 2 Raw data

Click here for additional data file.

Additional Information and Declarations

Competing Interests

Author Contributions

Animal Ethics

Data Availability

The authors declare there are no competing interests.

Ping Wang performed the experiments, analyzed the data, contributed reagents/materials/analysis tools, prepared figures and/or tables, approved the final draft.

Zengli Zhang prepared figures and/or tables.

Yujie Ma contributed reagents/materials/analysis tools.

Jun Lu and Shuiliang Wang performed the experiments.

Hu Zhao contributed reagents/materials/analysis tools, authored or reviewed drafts of the paper.

Jianming Tan conceived and designed the experiments, authored or reviewed drafts of the paper, approved the final draft.

Bingyan Li conceived and designed the experiments, analyzed the data, authored or reviewed drafts of the paper, approved the final draft.

The following information was supplied relating to ethical approvals (i.e., approving body and any reference numbers):

The animal protocol for this study was approved by the Institutional Animal Care and Use Committee (IACUC) of Fuzhou General Hospital (Fujian, China) and followed the regulatory animal care guidelines of the United State National Institute of Health (Bethesda, MD, USA).

The following information was supplied regarding data availability:

The raw measurements are available in the Supplemental Files.

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
