# Peer review of "Prognostic values of GMPS, PR, CD40, and p21 in ovarian cancer"

_PeerJ, doi:10.7717/peerj.6301_

## Round 0.1 · original submission · Major Revisions

Please address the comments of the reviewers and revise your manuscript appropriately

Reviewer 1 ·

Basic reporting

Please see pdf

Experimental design

Please see pdf

Validity of the findings

Please see pdf

Annotated reviews are not available for download in order to protect the identity of reviewers who chose to remain anonymous.

Reviewer 2 ·

Basic reporting

I have no concerns about the basic reporting of this article. It is easy to read, clear and well referenced

Experimental design

The conclusion of this study is widely exaggerated. I do not agree that you can go from Murine based model (why they didn’t use established ovarian cancer lines is not explained), compare the data to human data, identify one novel association (GMPS) and state you have a prognostic signature for OC. Post hoc ergo propter hoc it is not! This is highlighted by the fact in the discussion whereby authors outline that a human study came to a different conclusion then their work.

Prognostication should be done in the confines of already understood prognostic groups- i.e in HGS what gene are associated with better survival or in P53 mutants what passenger mutations allow for better survival. This paper doesn’t really have a hypothesis that can be actually tested. This paper asks: which mutations in(a selected –not sure how selected) MOSEC change during carcinogensis (as sequenced by ?means) and then which of these seem to have a human association in all ovarian cancers (a massive hetrogenous group) and then do any of these correlate to a survival difference (with no controlling of any confounding).

I think this paper limit it scope to simply – what are the driver mutations in MOSEC cell line? That would be within the means of this study and would make a contribution that would help with the application of this murine cancer model.

It isn’t clear how many genes/outputs were analyzed. However, if you there was more than 20 then an alpha off 0.05 is too permissive allowing associations by chance to enter into later analysis.

They refer to OC as a genetically homogenous entity – I.e. our markers prognosticate in OC … this is missing the work of the TCGA and the difference between histologies.

“Our experiments were in three biological replicates to reduce sample variations due to cell heterogeneous cultures” line 201 – so they have selected their sequenced population. Not clear how these cells were selected. Therefore, are these results generalizable to the MOSEC population, let alone human OC.

“In our current study, we found that GMPS was highly expressed in ovary cancer tissues compared with that of adjacent tissues and that GMPS expression was associated with RFS of patients”. – where did this normal tissue come from – it is not mentioned in the methods. Is this MOSEC-I vs MOSEC II?

I see nothing exciting about finding genes associated with cell proliferation are associated with cancer.

I do not see how GMPS is a better prognosticator than P53, histology, age, immunoscore etc

Validity of the findings

Apart from GMPS all the genes associated survival impact have been described in much better (human) studies.

There is no description of the NGS that was carried out. Was is exomic, genomic, intronic, panel etc

No mention of quality control, allele frequency cut offs, read depth etc

PIK3CA is associated with clear cell and endometrioid mainly – Is MOSEC associated with this histotype? I am still unclear what the MOSEC actual is a proxy of- it would seem it closely models HGS and therefore it is unusual to have a PIK3CA association

Additional comments

How does this in any way add to:
https://bcl.med.harvard.edu/wp-content/uploads/2013/10/Spentzos-et-al-2004.pdf

---

## Round 0.2 · Minor Revisions

Please respond to reviewer 2 comments referencing the methods section

Reviewer 2 ·

Basic reporting

I commend the authors, this has greatly improved since the first edition. The paper now reads well and the numerous typographical errors have been corrected.

My only comment would be that in the last paragraph of the introduction basically outlines the method. I think this would be better re-phrased so that is outlined the issue the paper seeks to address. I.e what is the applicability That is:

HGS has a terrible prognosis
DEGs could help better risk stratify women with the disease so they could receive the appropriate treatment
MOSECs are a credible model of HGS
Therefore DEGS examined in the MOSECs acting as a model of carcinogenesis and applied to the TCGA for prognostic significance.
The hypothesis is that DEGs identified in MOSEC cells could help yield molecular prognostic targets and explore the biology of HGS

Also in the discussion please add a specific paragraph for strengths followed by a separate paragraph exploring the weaknesses of your study. I believe there needs to be a little bit more introspection

Experimental design

The language of this paper has now very much improved.
The methods are much clearer. Previous concerns have been answered.

Validity of the findings

Again, I think we have to be very careful about the overstatement of this work. However I believe that the authors now have got the balance right- this is interesting discovery work and the use of the TCGA adds wait to the findings in MOSEC cells.

Additional comments

This paper is greatly improved. You should be congratulated.

---

## Round 0.3 · accepted · Accept

Well done you have made this manuscript very acceptable.

#